# Evaluation of *Larrea tridentata* Extracts and Their Antimicrobial Effects on Strains of Clinical Interest

**DOI:** 10.3390/ijms26031032

**Published:** 2025-01-25

**Authors:** Renata Morales-Márquez, Lucía Delgadillo-Ruiz, Alfredo Esparza-Orozco, Eladio Delgadillo-Ruiz, Rómulo Bañuelos-Valenzuela, Benjamín Valladares-Carranza, María Isabel Chávez-Ruvalcaba, Francisca Chávez-Ruvalcaba, Héctor Emmanuel Valtierra-Marín, Norma Angélica Gaytán-Saldaña, Marisa Mercado-Reyes, Luz Adriana Arias-Hernández

**Affiliations:** 1Academic Unit of Biological Sciences, Autonomous University of Zacatecas, Av. Preparatoria, Agronómica, Zacatecas 98068, Mexico; moralesrenata3a@gmail.com (R.M.-M.); alfredoesparzao@gmail.com (A.E.-O.); iasruv9si@uaz.edu.mx (M.I.C.-R.); gaytanangelica1@gmail.com (N.A.G.-S.); marisa.mercado@uaz.edu.mx (M.M.-R.); 2Department of Civil and Environmental Engineering, Engineering Division, University of Guanajuato, Av. Juárez, Col. Centro, Guanajuato 36000, Mexico; e.delgadillo@ugto.mx (E.D.-R.); arhadriana@ugto.mx (L.A.A.-H.); 3Academic Unit of Veterinary Medicine and Animal Husbandry, Autonomous University of Zacatecas, Kilómetro 31.5 Carretera Zacatecas-Fresnillo, Enrique Estrada, Zacatecas 98502, Mexico; banuelosluis257@gmail.com; 4Center for Research and Advanced Studies in Animal Health, Autonomous University of the State of Mexico, Toluca 50200, Mexico; bvalladaresc@uaemex.mx; 5Academic Unit of Nutrition, Autonomous University of Zacatecas, Carretera Zacatecas-Guadalajara, Km 6, Ejido la Escondida, Zacatecas 98160, Mexico; charuva@uaz.edu.mx; 6Academic Unit of Agronomy, Autonomous University of Zacatecas, Carretera Federal 54 Km 15.5, Zacatecas 98170, Mexico; hectorv@uaz.edu.mx

**Keywords:** antimicrobial activity, chemical composition, creosote bush, extracts

## Abstract

The use of medicinal plants represents an alternative method for bacterial control due to their chemical compositions. This study’s objective was to determine the inhibitory capacity of *Larrea tridentata* extracts against microbial strains of clinical interest. Four extracts were prepared, their phytochemical profiles were determined, and their antioxidant capacities were quantified. Additionally, the minimum concentrations of hemolysis were determined using human blood erythrocytes. For the extracts’ growth inhibitory capacity, six bacterial and two fungal strains were evaluated using the disk diffusion test. Commercial medications specific for each strain were used as controls. The ethanolic extracts registered the greatest diversity of metabolites related to antibacterial activity. The inhibitory activities of the ethanolic extract and the Cedax^®^ control were similar for *Enterococcus faecalis*. A principal component analysis was performed with X^2^ and ANOVA tests to identify the relationships and the effects of the extracts on bacterial inhibition, obtaining *p* > 0.05 with a confidence level of 95%. This research highlights the potential of *L. tridentata* extracts as an alternative treatment and to mitigate the growing problem of resistance to traditional antibiotics.

## 1. Introduction

In recent years, drug resistance has become the biggest problem affecting healthcare systems [1]. The inadequate use of antimicrobial drugs has generated a worldwide problem of drug resistance [2]. Pathogenic microorganisms such as fungi and bacteria have developed resistance to multiple antimicrobial agents [3]. Aiming to address this problem, the identification of new molecules from plant extracts for use as antimicrobial control is becoming a key topic of growing interest [4,5].

The World Health Organization (WHO) recognizes the value of plants and estimates that 80% of the world’s population relies on them for alternative and traditional medicine in primary healthcare [6,7]. Over the last 35 years, 70% of high-end drugs used in cancer and infectious disease treatments have been developed based on plants [7,8]. For example, antimicrobial compounds from plants are generally obtained from essential oils, oleoresins, and extracts obtained from flowers, bulbs, rhizomes, bark, and fruits [9]. These compounds are rich in molecules such as alkaloids, terpenoids, phenols, flavonoids, tannins, and polysaccharides [10,11], called secondary metabolites, which are synthesized by the secondary metabolism of plants in response to adverse abiotic or biotic conditions [12,13]. Plant-derived secondary metabolites have shown significant antioxidant, antibacterial, and antifungal benefits, making them notable for their inhibitory potential against microorganisms [14]. 

In plant species such as *Larrea tridentata* (DC.) Coult, commonly called creosote bush, around 50% of the dry weight of the leaves corresponds to extractable secondary metabolites such as flavonoids, saponins, essential oils, and antioxidants [15,16,17]. Historically, *L. tridentata* has been used in traditional medicine against approximately 50 diseases, mainly renal and hepatic diseases [17]. The antioxidant capacity of *L. tridentata* promotes iron reduction, as observed using the FRAP (ferric reducing antioxidant power) and DPPH (2,2-diphenyl-1-picryl-hydrazyl-hydrate) methods, which are used to evaluate the free radical-scavenging capacity associated with phenolic compounds and nordihydroguaiaretic acid (NDGA) [18]. In fact, evidence suggests that NDGA promotes neurogenesis and angiogenesis in rats with cerebral ischemia [19].

In the last decade, research has focused on describing new chemical compounds from *L. tridentata* [16]. For example, Jitsuno and Mimaki [20] described thirteen new metabolites of the glucosylated triterpene group, and Yokosuka et al. [21] identified two new metabolites of the glucosylated lignan group named larrealignans A and B. These authors suggest that the cytotoxic activity of these metabolites against HL-60 human promyelocytic leukemia cells partially explains the alternative use of *L. tridentata* to treat cancers. Metabolites in extracts from *L. tridentata* have shown high chemical and cellular activity as antibiotic or antimicrobial agents against bacteria, fungi, and even parasitic animals, such as protozoa [16]. In the literature, many methods have been reported for extracting secondary metabolites from *L. tridentata* as alcoholic or water-based extracts [16,22]. For example, dichloromethane extracts inhibit 96.5% of β-(1,3)-glucanase’s enzymatic activity, contrasting with raw methanol extracts [23]; this enzyme is involved in plant–fungal pathogen interactions [22]. Other authors found that the antioxidant activity of an ethanol–water (60:40) extract had the most efficient antioxidant properties compared to ethanol or water extracts [24].

Moreover, the antibacterial potential of lignans and flavonoids has been shown at low concentrations (12.5 to 50 µg/mL), which are effective against gram-negative strains (*Stenotrophomona maltophilia*, *Escherichia coli*, *Acinobacter baumannii*, *Haemophilus influenzae*, *Pseudomonas aeruginosa*, *Klebsiella pneumoniae*, and *Enterobacter cloacae*) and gram-positive bacteria (*Staphylococcus aureus*, *Streptococcus pneumoniae*, *Listeria monocytogenes*, and *Enterococcus faecalis*) [25]. However, studies evaluating the inhibitory capacity of the secondary metabolites of *L. tridentata* against pathogenic microorganisms, such as fungi and bacteria, through different extraction methods are limited. In this study, we evaluated the inhibitory, antioxidant, and hemolytic activity of *L. tridentata* extracts, obtained using four extraction methods, against two fungal and six bacterial strains. The use of secondary metabolites from plant extracts may offer the most promising method for addressing antimicrobial resistance to drugs.

## 2. Results

### 2.1. Phytochemical Characterization

According to phytochemical tests, ultrasound-assisted extraction (UAE) showed the highest presence of metabolites, followed by ethanolic extraction (EE), which recorded the presence of anthocyanins, coumarins, lactones, tannins, and saponins (Table 1). Conversely, UAE and EE were negative for quinones. Moreover, decoction extraction (DE) was negative for sesquiterpenlactones, bic saponins, and flavonoids (2) (Table 1).

Two principal components (PCs) from the principal component analysis (PCA) explained 83% of the variation in metabolites based on the extraction method used. The first PC accounted for 68% of the variation, while the second PC explained 15% (Figure 1). Flavonoids (2), Salk saponins, esters, and sesquiterpenlactones were related to the EE method, whereas anthocyanins, carbohydrates, and quinones were linked to the DE method (Figure 1). Conversely, other metabolites, such as tannins, coumarins, and alkenes, were not strongly associated with any specific extraction method (Figure 1).

### 2.2. Antioxidant Characterization

The results of the antioxidant capacity tests for DPPH (2,2-difenil-1-picril-hidrazil-hidrato) and FRAP (ferric-reducing antioxidant power), expressed as the antioxidant capacity of Trolox (TE; μM TE/g), are shown in Table 2. Among the four extracts, EE revealed the highest antioxidant capacity. For flavonoids, EE recorded the highest concentration (7.83 mg/mL); for phenols, UAE and EE recorded similar concentrations; and for tannins, UAE showed the highest concentrations (0.94 mg/mL).

### 2.3. Hemolysis

Table 3 shows the results of hemolysis assays, indicating the total number of positive effects of the four extracts by exposure time and dilution. At a 1:10 dilution, six positive tests were recorded during the first hour, whereas at a 1:100 dilution, six positive tests were observed during the third hour.

Figure 2 shows the positive results of the hemolysis assays according to the dilution factor (Figure 2a) and the exposure time to the extracts (Figure 2b). EE recorded the highest number of positive tests at 1:10,000 dilutions and showed the highest hemolytic activity across all dilutions. The four extracts exhibited hemolytic activity at a 1:10 dilution, but their effects gradually decreased with higher dilutions (Figure 2a). Regarding exposure time, EE caused hemolysis in all observation periods, while UAE maintained its hemolytic effect for up to 3 h. In contrast, EE and decoction extraction (DE) were positive for hemolysis only up to 3 and 24 h of exposure, respectively (Figure 2b).

### 2.4. Microbial Inhibition

The lengths of the inhibitory halos on the microbial strains were the longest for EE, followed by UAE (Figure 3). EE showed the highest inhibition averages against *Enterococcus faecalis*, with values of 5.68, 4.56, and 4.32 mm (Figure 3). Against *Streptococcus* sp., EE recorded averages of 3.31, 2.48 mm, and 4.27 mm. UAE showed high inhibition averages against *Saccharomyces cerevisiae*, with values of 4.04, 2.96, and 1.99 mm. Conversely, DE and EE exhibited the lowest inhibitory effects (Figure 3).

The inhibitory effects of the four extracts on the microbial strains were not significantly different (*p* > 0.05). The means (M) and standard deviations (SD) of the inhibitory effects of UAE on *Candida albicans* were similar to those of DE (M = 1.25, SD = 0.81 and M = 0.89, SD = 1.41, respectively). Overall, EE demonstrated better inhibitory effects than UAE. However, for *Staphylococcus epidermidis*, the effects of EE and UAE were comparable (M = 2.72, SD = 2.92 and M = 2.11, SD = 1.51, respectively; *p* = 0.97). Similarly, the inhibitory effects of EE and the control drug Cedax^®^ on *Enterococcus faecalis* were not significantly different (M = 4.85, SD = 1.81 and M = 6.3, SD = 0.82, respectively; *p* = 0.17).

## 3. Discussion

Secondary metabolites are chemical compounds with a low molecular weight and diverse chemistry, and they have been used as medical alternatives. The phytochemical characterization of plants is essential to obtain more information. In the first trial of this research, flavonoids, saponins, and sterols were found to be present in the 70% ethanolic solution, in addition to carbohydrates, quinones, unsaturations, and metabolites, as also reported by Bañuelos et al. [26]. Likewise, the presence of tannins was found to have a relevant relationship with ethanolic extraction. It has been documented that tannins are the main agents in the inhibition of bacterial activity, resulting in a reduction in methanogenesis in ruminants [27].

The results of this research suggest that phenolic compounds were present in abundance; such compounds are responsible for the antioxidant capacity. Morales–Ubaldo et al. [16] documented the main antioxidant compounds abundant in the plant kingdom and present in *Larrea tridentata* using the ferric reducing power/antioxidant (FRAP) and 2,2-diphenyl-2-picrylhydrazyl radical-scavenging (DPPH) methods, as in this work.

Regarding methodologies and extract types, 70% ethanol, an extractant solution (methanol–water–formic acid: 80:18:2), infusion, and boiling were used, yielding results similar to those obtained with an ethanolic–water solution (60:40) in a study by Skouta et al. [24]; these authors obtained efficient results in terms of antioxidant properties with values reported in percentages, similar to this research, with DPPH and FRAP used as the antioxidant tests and methodologies. Both investigations observed that the best performance was obtained with the 70% ethanolic solution.

The hemolytic effect was evaluated on the basis of negative and positive controls. The evaluation was determined according to positive or negative hemolytic activity based on observation time and dilution. All four extraction types were positive at a 1:10 dilution, with results similar to those reported by Silva et al. [28]. Hemolytic activity is due to phenolic compounds and flavonoids, both of which were found to be present in the extracts in this research; these compounds also produce echinocytes, products of antioxidant action, as reported by López et al. [29]. In this work, significant results (*p* < 0.05) were obtained for bacterial inhibition in relation to the strain *Candida albicans*; it was observed that UAE had the same behavior as the infusion solution. For *Staphylococcus epidermidis*, UAE and EE had the same effect, being that for all the tests, they were different.

In these same results, a relevant relationship was obtained according to the efficiency of the 70% ethanolic solution, where it acted in the same way as the control drug Cedax^®^, with both treatments producing equal inhibition halo lengths. Similar results were also obtained by Mendez et al. [30], who evaluated water and ethanol extracts of *Larrea tridentata* leaves and showed that the ethanolic extract was the most efficient in inhibiting the growth of *Escherichia coli* and *Staphylococcus aureus*. Similarly, Turner et al. [31] determined that the ethanolic extract was the most efficient in terms of bactericidal activity against *Staphylococcus aureus*, *Streptococcus pyogenes*, *Bacillus cereus*, *Escherichia coli*, and *Pseudomonas aeruginosa*; however, they did not show results for *Enterococcus faecalis*. Likewise, Montemayor et al. [32] determined that the efficiency of the ethanolic extract did not depend on ethanol but on the chemical activity of the extract itself, with results obtained from tests with strains such as *Staphylococcus aureus* and *Escherichia coli*.

## 4. Materials and Methods

### 4.1. Plant Collection

Fresh leaves and flowers of *Larrea tridentata* were collected during the summer season, prior to the rains, in Villa de Cos, Zacatecas, Mexico (22°58′ to 24°01′ N latitude, 101°28′ to 102°44′ W longitude; altitude: 1800–2800 m). The fresh plant material was washed with water, dried, and ground into a fine powder using a grinder. The powdered material was stored in plastic bags in a refrigerator and subsequently used for extraction.

### 4.2. Extraction Procedure

Four different extraction methods were employed to obtain extracts of *L. tridentata*:

Decoction extraction (DE): 50 g of plant material was boiled in 400 mL of water for 15 min, filtered, and stored at 4 °C.

Ethanolic extraction (EE) [33,34]: Samples were prepared in 70% ethanol (J.T. Baker^®^, Radnor, PA, USA), with 12.5 g of plant material per 100 mL of solvent. The samples were macerated for 30 days in amber jars at room temperature. After maceration, the supernatant was recovered via filtration and stored at room temperature.

Infusion extraction (IE): 50 g of plant material was mixed with 400 mL of boiling water for 15 min. The sample was then filtered and stored at 4 °C.

Ultrasound-assisted extraction (UAE): A solution of methanol (J.T. Baker^®^, Radnor, PA, USA), water, and formic acid (Sigma-Aldrich, St. Louis, MO, USA) (80:18:2 ratio) was prepared. Then, 3 g of plant material was added to 30 mL of the solution. The mixture was stirred for 30 s and subjected to ultrasound in five one-minute cycles at 60 Hz, maintaining a temperature of 4 °C between cycles. The extracts were macerated for 30 days in amber jars at room temperature. After maceration, the supernatant was recovered via filtration and stored at room temperature.

### 4.3. Qualitative Tests for Chemical Profile

Phytochemical tests were performed to identify the presence of metabolites following the methodologies of Domínguez [35], Bañuelos-Valenzuela et al. [26], and Rivero et al. [36] (Table 4). The results were compiled into a presence/absence matrix, with rows representing the samples and columns corresponding to the metabolite types.

### 4.4. Determination of pH

The pH of the samples was measured using a Thermo Scientific^®^ Orion Star A211 potentiometer [37].

### 4.5. Quantitative Tests for Chemical Profile

The total phenol content was measured using a Folin–Ciocalteu assay [38]. A 50 µL extract aliquot was added to an ELISA microplate, followed by 250 µL of Folin–Ciocalteu reagent (Sigma-Aldrich, St. Louis, MO, USA) diluted 10× with water. After a 2 min reaction in the dark, 200 µL of Na_2_CO_3_ (Sigma-Aldrich, St. Louis, MO, USA) (75 mg/L) was added. The plate was incubated in the dark for 2 h, and absorbance was measured at 765 nm with an ELISA microplate reader (Thermo Scientific) using a gallic acid (Sigma-Aldrich, St. Louis, MO, USA) (in methanol 80%) calibration curve. The results are expressed in mg/mL.

We measured the total flavonoid content using the method developed by Zhishen et al. [39]. A volume of 100 µL of the extract was placed in a microcentrifuge tube, followed by 40 µL of distilled water and 30 µL of NaNO_3_ (5%) (Sigma-Aldrich, St. Louis, MO, USA). The mixture was left to react for five minutes. Then, 30 µL of AlCl_3_ (10%) (Sigma-Aldrich, St. Louis, MO, USA) was added, and the mixture was left to react for six minutes. Next, 200 µL of NaOH (1 N) (Sigma-Aldrich, St. Louis, MO, USA) was added, followed by 240 µL of distilled water. From this mixture, 300 µL was transferred to an ELISA microplate. Absorbance was measured at 510 nm using an ELISA microplate reader (Thermo Scientific, Waltham, Massachusetts, USA) and compared to a catechin (in 80% methanol) calibration curve. The results are expressed in mg/mL.

We measured condensed tannins using the method of Amarowicz and Pegg [40]. A volume of 50 µL of the extract was placed on an ELISA microplate, followed by 250 µL of a solution consisting of HCl (J.T. Baker^®^, Radnor, PA, USA) (8% in methanol) and vanillin (Sigma-Aldrich, St. Louis, MO, USA) (1% in methanol) in a 1:1 ratio. Absorbance was measured at 492 nm using an ELISA microplate reader (Thermo Scientific, Waltham, Massachusetts, USA) and compared to a catechin calibration curve (in methanol). The results are expressed in mg/mL. All the above-mentioned tests were performed in triplicate.

We carried out the following two assays to measure the antioxidant capacity: the ferric reducing/antioxidant power (FRAP) and the 1,1-diphenyl-2-picrylhydrazyl (DPPH) (Sigma-Aldrich, St. Louis, MO, USA) assays.

The FRAP assay was carried out according to the methodology described by Benzie and Strain [41] and modified by Álvarez et al. [42]. We prepared a solution of a working FRAP reagent, with acetate buffer (0.3 M, pH 3.6), TPTZ (2,4,6-tripyridyl-S-triazine) (Sigma-Aldrich, St. Louis, MO, USA) solution (10 mM in 40 mM HCl), and FeCl_3_ (20 mM) (Sigma-Aldrich, St. Louis, MO, USA) in a 10:1:1 ratio. A volume of 24 µL of extract was placed on an ELISA microplate, followed by 180 µL of the working FRAP reagent. Absorbance was measured at 630 nm using an ELISA microplate reader (Thermo Scientific). We used Trolox to make a standard, and the results are expressed as Trolox equivalent (TE) (μM TE/g).

The antioxidant capacity obtained from the DPPH assay was measured according to the methodology by Brand–Williams et al. [43] and Chew et al. [44]. A solution of 1 mM DPPH in methanol was stirred. The absorbance of the solution was adjusted to 1 (<0.010) at 480 nm. Then, 20 µL of extract was placed on an ELISA microplate, followed by 280 µL of DPPH solution (Sigma-Aldrich, St. Louis, MO, USA), and incubated for 30 min in the dark covered with aluminum foil. Then, decreasing absorbance was monitored at 480 nm. A calibration curve was plotted using standard solutions of Trolox–methanol with concentrations in the range of 0 to 0.0008 µM, and the results are expressed as Trolox equivalent (TE) (μM TE/g).

### 4.6. Determination of Hemolysis

We collected 10 mL of human blood in a heparinized tube. The tube containing non-coagulated blood was centrifuged at 2500 rpm for 10 min, and the serum fraction was carefully removed. Erythrocytes were washed three times with a solution of PBS (50% *v*/*v*) and glucose (2.25% *w*/*v*), with centrifugation after each wash at 2500 rpm for 10 min at 10 °C. The erythrocytes were resuspended at a concentration of 0.1% (*v*/*v*) in a suspension regulator containing PBS (BIO-RAD, Berkeley, CA, USA) (50% *v*/*v*), glucose solution (Sigma-Aldrich, St. Louis, MO, USA) (2.25% *w*/*v*), and gelatin (0.05% *w*/*v*) [28].

For the hemolysis assay, 100 µL of the 1% erythrocyte suspension was added to an ELISA microplate, followed by 100 µL of the extract. Serial dilutions of the extract were tested at ratios of 1:10, 1:100, 1:1000, 1:10,000, 1:100,000, and 1:1,000,000. The negative control consisted of 100 µL of the 1% erythrocyte suspension alone, while the positive control consisted of 100 µL of the 1% erythrocyte suspension mixed with 100 µL of 1% Triton X (BIO-RAD, Berkeley, CA, USA). The erythrocytes were exposed to the extracts, and their activity was recorded at 1, 2, 3, and 24 h of exposure. The results were categorized as positive or negative through a comparison with the positive and negative controls.

### 4.7. Microbial Strains

In this study, eight strains were used, including six bacterial and two fungal isolates (Table 1). The strains were obtained from the Microbiology Laboratory of the Academic Unit of Chemical Sciences of the Autonomous University of Zacatecas (Mexico). Their identification was performed using biochemical tests in commercial panels with the Phoenix 100 system (Becton Dickinson and Company^®^, Sparks, MD, USA) by the Bacteriology Laboratory of the Hospital General in Fresnillo, Zacatecas, Mexico.

The strains were preserved in nutrient media with glycerol 60% stocks stored at −70 °C. Prior to antibacterial testing, the microorganisms were reactivated in nutrient broth and Mueller–Hinton agar (DIBICO, Proquisur, Mexico City, Mexico) at 37 °C in a Thermo Scientific (Waltham, MA, USA) incubator for 24 h [45].

### 4.8. Antimicrobial Evaluation

We examined the antimicrobial activity of the *Larrea tridentata* extracts by using the Kirby–Bauer disk diffusion test. Bacteria and fungi colonies from fresh cultures were suspended in tubes containing sterile saline solution, and the turbidity of the suspensions was adjusted to match the 0.5 McFarland standard. A 100 µL aliquot of each bacterial or fungal suspension was evenly spread onto the surface of Mueller–Hinton (DIBICO, Proquisur, Mexico City, Mexico) agar plates using a sterile loop. Disks impregnated with *Larrea tridentata* extracts were carefully placed at four distinct points on the agar surface using sterile forceps [45]. Adequate spacing between the disks was maintained to prevent overlapping zones of inhibition. The inoculated plates were incubated at 37 °C for 24 h. After incubation, the diameters of the inhibition zones around each disk were measured in millimeters using a ruler or caliper [46,47,48]. The negative controls included sterile distilled water, 70% ethanol, and methanol, while the positive controls consisted of specific drugs selected according to the type of bacteria or fungi being tested (Table 5). Each assay was performed in duplicate to ensure reproducibility.

### 4.9. Multivariate and Statistical Analyses

To identify the relationships between the metabolites of the *Larrea tridentata* extracts and the extraction method used, a principal component analysis (PCA) was performed. For this purpose, a presence/absence matrix of metabolites for each extraction method was created. Subsequently, matrix transformation was applied using the Chi-square method (X^2^), with the “decostand” function of the “vegan” package [49]. This method divides the row sums and the square root of the column sums and makes adjustments using the square root of the matrix total [50]. This transformation is appropriate for the multivariate ordination of binary data [51]. To generate the PCA, we used the “PCA” function of the “FactoMineR” package [52], and matrix scaling was applied to a mean of zero (μ = 0) and a standard deviation of one (σ = 1). To determine the effects of the extracts on bacterial inhibition, an analysis of variance (ANOVA) and Tukey’s post hoc test were performed to identify differences between groups. Statistical tests were evaluated with α = 0.05. All data analyses were performed with the free software R version 4.2.3 (Vienna, Austria) [53].

## 5. Conclusions

In each of the tests carried out, the 70% ethanolic extract had the highest presence of metabolites, which influenced its bacterial inhibition capacity. Additionally, this 70% ethanolic extract obtained the best average microbial inhibition halo, showing an effect equal to that of the control drug Cedax^®^. Thus, it is concluded that the 70% ethanolic extract is the best option for application in treatments in future trials. Future studies should emphasize its chemical aspects by examining the concentrations of each important compound, such as flavonoids, tannins, saponins, phenols, and anthocyanins. Future studies should also determine which molecules play an important role in order to use them for specific treatments and prevent cytotoxic effects.

## Figures and Tables

**Figure 1 ijms-26-01032-f001:**
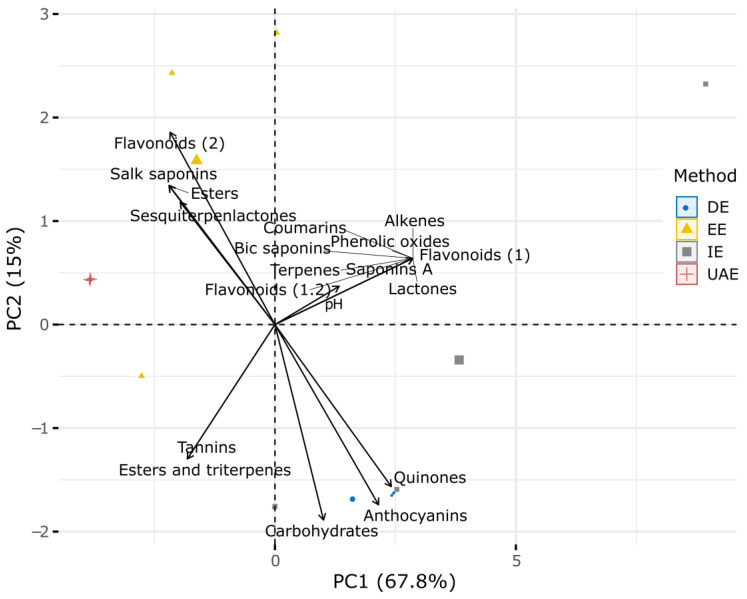
Principal component analysis (PCA) biplot of secondary metabolites in relation to extraction methods, namely, ethanolic extraction (EE), ultrasound-assisted extraction (UAE), infusion extraction (IE), and decoction extraction (DE), and their pH values. The percentages of variance explained for the first and second principal components (PC1 and PC2, respectively) are shown.

**Figure 2 ijms-26-01032-f002:**
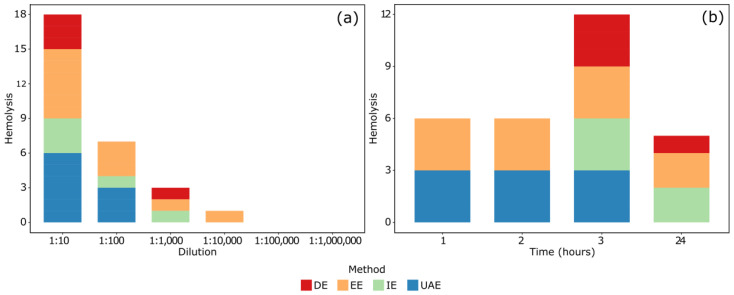
Hemolytic capacity of extracts according to (**a**) dilution and (**b**) exposure time. Ethanolic extraction (EE), ultrasound-assisted extraction (UAE), infusion extraction (IE), decoction extraction (DE).

**Figure 3 ijms-26-01032-f003:**
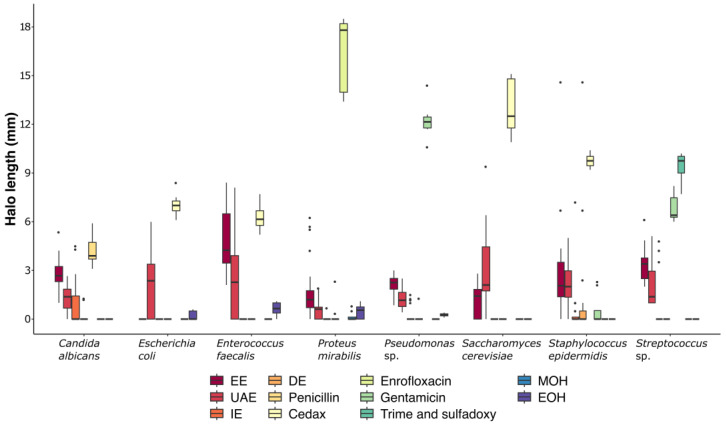
Boxplot for the effects of the extracts (ethanolic extraction (EE), ultrasound-assisted extraction (UAE), infusion extraction (IE), and decoction extraction (DE)) and the controls (penicillin, Cedax^®^, enrofloxacin, gentamicin, Trime and sulfadoxy, MOH (methanol), and EOH (ethanol 70%)) against eight microbial strains (*Candida albicans*, *Escherichia coli*, *Enterococcus fecalis*, *Proteus mirabilis*, *Pseudomonas* sp., *Saccharomyces cerevisiae*, *Staphylococcus epidermidis*, and *Streptococcus* sp.). Each box represents the variance, mean, and standard deviation of the inhibition halo for each treatment. Black dots representing observations outside the 9–91 percentile range.

**Table 1 ijms-26-01032-t001:** Phytochemical profiles of extracts based on the presence/absence of metabolites.

Metabolites Detected	Extraction Methods			
	UAE	EE	DE	IE
Lactones	+	+	+	+
Cumarins	+	+	+	+
Carbohydrates	+	−	+	+
Esters	+	+	+	+
Sesquiterpenlactones	+	−	−	−
Flavonoids (1)	+	+	+	+
Insaturations	+	+	+	+
Phenolic oxides (vegetable tannins)	+	+	+	+
Saponins A	+	+	+	+
Salk saponins	+	+	+	+
Bic saponins	+	+	−	−
Flavonoids (H_2_SO_4_ test) (1.2)	+	+	+	+
Salk esters and triterpenes	+	+	+	−
Flavonoids (Shinoda test) (2)	+	+	−	−
Quinones	−	−	+	+
Terpenes	+	+	+	+
Anthocyanins	−	−	+	+
Tannins	+	+	+	+

+: presence; −: absence. A: agitation. Salk: Salkowski. Bic: sodium bicarbonate. Ethanolic extraction (EE), ultrasound-assisted extraction (UAE), infusion extraction (IE), decoction extraction (DE). Salk: Salkowski.

**Table 2 ijms-26-01032-t002:** Antioxidant capacity of the extracts according to DPPH and FRAP assays of TE (μM TE/g) and flavonoid, phenol, and tannin quantifications expressed in mg/mL.

Extracts	DPPH	FRAP	Flavonoids	Phenols	Tannins
	TE	TE	mg/mL	mg/mL	mg/mL
UAE	0.36 ± 0.12	2.78 ± 0.24	4.48 ± 0.68	3.69 ± 0.34	0.94 ± 0.18
EE	0.50 ± 0.04	2.63 ± 0.55	6.88 ± 0.95	3.89 ± 0.02	0.78 ± 0.18
DE	0.98 ± 0.42	2.91 ± 0.20	2.63 ± 0.24	2.83 ± 0.55	0.20 ± 0.08
IE	0.67 ± 0.08	2.83 ± 0.14	1.83 ± 0.14	2.68 ± 0.90	0.06 ± 0.09

Ethanolic extraction (EE), ultrasound-assisted extraction (UAE), infusion extraction (IE), decoction extraction (DE), Trolox equivalent (TE).

**Table 3 ijms-26-01032-t003:** Hemolysis assays according to exposure time of the erythrocytes to the extracts.

Time (h)	1:10	1:100	1:1,000	1:10,000	1:100,000	1:1,000,000	Total
1	6	0	0	0	0	0	6
2	6	0	0	0	0	0	6
3	6	6	0	0	0	0	12
24	6	7	3	1	0	0	5
Total	6	7	3	1	0	0	29

**Table 4 ijms-26-01032-t004:** Qualitative tests used to determine the chemical profiles of *Larrea tridentata* extracts.

Compound Chemist	Test	Procedure
*Insaturations*	KMnO_4_	To 100 µL of the extract, three drops of 2% KMnO_4_ (Sigma-Aldrich, St. Louis, MO, USA) in water were added dropwise. A positive test was indicated by discoloration or the formation of a brown precipitate (manganese dioxide).
*Phenolic oxydryls (vegetable tannins)*	FeCl_3_	To 100 µL of the extract, three drops of 12.5% FeCl_3_ (Sigma-Aldrich, St. Louis, MO, USA) in water were added. A positive test was indicated by the formation of a red, blue–violet, or green precipitate.
*Carbonyl groups*	2–4 Dinitxophenylhydrazine	One drop of 2,4-dinitrophenylhydrazine solution in 6N HCl (J.T. Baker^®^, Radnor, PA, USA) was added to 100 µL of the extract. A positive test was indicated by a yellow or orange precipitate.
*Triterpenes and sterols*	Liebermann–Burchard	The reagent was prepared by mixing 1 mL CH_3_COOH (J.T. Baker^®^, Radnor, PA, USA), 1 mL CHCl_3_ (J.T. Baker^®^, Radnor, PA, USA), and 1 drop of H_2_SO_4_ (J.T. Baker^®^, Radnor, PA, USA), followed by cooling to 0 °C. To 100 µL of the extract, three drops of the reagent were added. Positive results were observed as blue, green, red, or orange colors over time.
	Salkowski	To 200 µL of the extract, 500 µL of H_2_SO_4_ (J.T. Baker^®^, Radnor, PA, USA) was added. A positive test was indicated by yellow or red coloration, confirming the presence of sterols or methylsterols.
*Carbohydrates*	Molish	Two drops of Molisch’s reagent were added to 100 µL of the extract, followed by 500 µL of H_2_SO_4_ (J.T. Baker^®^, Radnor, PA, USA). A positive test was indicated by the presence of a purple ring at the interface.
	Coumarin	To 100 µL of the extract, 100 µL of 10% NaOH (Sigma-Aldrich, St. Louis, MO, USA) was added. A positive test was indicated by yellow coloration, which disappeared upon acidification with HCl (J.T. Baker^®^, Radnor, PA, USA).
	Lactone	To 100 µL of the extract, 100 µL of a 10% NaOH (Sigma-Aldrich, St. Louis, MO, USA) alcoholic solution was added. A positive test was indicated by yellow or orange coloration that disappeared after the addition of a few drops of HCl.
*Sesquitexpenlactones*	Baljet	To 100 µL of the extract, 3–4 drops of Baljet solution [10 mL of C_6_H_3_N_3_O_7_ 1% (Sigma-Aldrich, St. Louis, MO, USA), 10 mL NaOH 10% (Sigma-Aldrich, St. Louis, MO, USA)] mixed solution were added. A positive test was indicated by the appearance of orange or dark coloration.
*Flavonoids*	H_2_SO_4_	To 100 µL of the extract, 500 µL of H_2_SO_4_ (J.T. Baker^®^, Radnor, PA, USA) was added. Positive results were indicated by yellow coloration for flavonoids, orange–cherry for flavones, red–blue for chalcones, and red–purple for quinones.
	Shinoda	100 µL of the extract was mixed with 100 µL of ethanol, followed by the addition of 0.1 g magnesium filings (J.T. Baker^®^, Radnor, PA, USA). The sample was boiled, and three drops of concentrated HCl (J.T. Baker^®^, Radnor, PA, USA) were added. Positive results were indicated by orange, red, pink, blue, or violet coloration.
*Alkaloids*	Dragendorf	Two to three drops of reagent A (bismuth nitrate (Sigma-Aldrich) and glacial acetic acid) and reagent B (potassium iodide (Sigma-Aldrich)) were added to 100 µL of the extract. A positive test was indicated by orange to reddish coloration.
	Mayer	To 2 mL of the extract, 4 mL of Mayer’s reagent [Mercury and potassium iodide (Ricca Chemical^®^, Arlington, TX, USA)] was added. A positive test was indicated by yellow coloration with precipitates.
*Saponins*	Agitation	1 mL of the extract was dissolved with 1 mL of water in a test tube and shaken vigorously for 3–5 min. A positive test was indicated by stable foam with a honeycomb appearance for 30 min.
	NaHCO_3_	To 100 µL of the extract, 2–3 drops of H_2_SO_4_ (J.T. Baker^®^, Radnor, PA, USA) were added, and the mixture was lightly shaken. Then, 2–3 drops of 10% NaHCO_3_ (J.T. Baker^®^, Radnor, PA, USA) solution were added. A positive test was indicated by bubbles that persisted for over 1 min.
	Salkowski	To 100 µL of the extract, 100 µL of CHCl_3_ (J.T. Baker^®^, Radnor, PA, USA) was added, followed by 100 µL of H_2_SO_4_ (J.T. Baker^®^, Radnor, PA, USA). A positive test was indicated by the appearance of a red color.
*Aromaticity*	H_2_SO_4_-CH_2_O	One drop of a mixture of H_2_SO_4_ (J.T. Baker^®^, Radnor, PA, USA) and CH_2_O (J.T. Baker^®^, Radnor, PA, USA) was added to 100 µL of the extract dissolved in a non-aromatic solvent. A positive test was indicated by red or violet coloration.
*Anthocyanins*	HCl	To 1 mL of the extract, 5 mL of 10% HCl (J.T. Baker^®^, Radnor, PA, USA) was added, and the mixture was boiled in a water bath. A positive test was indicated by pale pink coloration.
*Terpenoids*	H_2_SO_4_	To 5 mL of the extract, 4 mL of chloroform and 4 mL of (J.T. Baker^®^, Radnor, PA, USA) were added. A positive test was indicated by reddish–brown coloration at the interface.
*Tannins*	FeCl_3_	1 mL of the extract was boiled, and 20 mL of water was added, followed by three drops of 0.1% FeCl_3_ (Sigma-Aldrich, St. Louis, MO, USA). A positive test was indicated by green or blue coloration.
*Steroids*	H_2_SO_4_	To 2 mL of CH_3_COOH (J.T. Baker^®^, Radnor, PA, USA), 0.5 mL of the extract was added, followed by 2 mL of H_2_SO_4_ (J.T. Baker^®^, Radnor, PA, USA). A positive test was indicated by violet, blue, or green coloration.

**Table 5 ijms-26-01032-t005:** Microbial strains used in resistance assays to *Larrea tridentata* extracts and commercial drugs (control drugs).

Strain	Type	Control Drugs
** *Candida albicans* **	Yeast	Penicillin (Pharmalife®, Veritrade, Mexico City, Mexico)
** *Escherichia coli* **	Gram-negative	Cedax^®^ (Sanfer®, Laboratorios Sanfer, Mexico City, Mexico)
** *Enterococcus faecalis* **	Gram-positive	Cedax (Sanfer®,Laboratorios Sanfer, Mexico City, Mexico)
** *Proteus mirabilis* **	Gram-negative	Enflofloxacin (ALTIA®, PYMES de BME, Spain)
***Pseudomonas*** **sp.**	Gram-negative	Gentamine (Halvet®, Veritrade, Mexico City, Mexico)
** *Saccharomyces cerevisiae* **	Yeast	Cedax^®^ (Sanfer®, Laboratorios Sanfer, Mexico City, Mexico)
** *Staphylococcus epidermidis* **	Gram-positive	Cedax^®^ (Sanfer®, Laboratorios Sanfer, Mexico City, Mexico)
***Streptococcus*** **sp.**	Gram-positive	Gentamine and Trimethylsulfadoxy (Novag®, Fressines, France)

## Data Availability

Data will be made available on request.

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
