# Peer review of "Evaluation of Larrea tridentata Extracts and Their Antimicrobial Effects on Strains of Clinical Interest"

_ijms, 2025, doi:10.3390/ijms26031032_

Round 1
Reviewer 1 Report
Comments and Suggestions for Authors
In sections 2.4 Methods and 4.7 Results, reference is made exclusively to bacteria, despite the inclusion of microorganisms belonging to the fungal kingdom, such as Candida albicans and Saccharomyces cerevisiae. To ensure accuracy and consistency with the content, it would be appropriate to explicitly mention fungi in these sections. Furthermore, it is recommended to modify the titles of both sections to adequately reflect the presence of microorganisms of different nature.
Author Response
Responses
Reviewer #1
1.-In sections 2.4 Methods and 4.7 Results, reference is made exclusively to bacteria, despite the inclusion of microorganisms belonging to the fungal kingdom, such as Candida albican and Saccharomyces cerevisiae. To ensure accuracy and consistency with the content, it would be appropriate to explicitly mention fungi in these sections. Furthermore, it is recommended to modify the titles of both sections to adequately reflect the presence of microorganisms of different nature.
Response: In the 'Methods' and 'Results' sections, we explicitly used terms such as 'Microbial inhibition,' 'Microbial strains,' and other 'microbial' references to denote fungi and bacteria, ensuring accuracy and consistency throughout the content as suggested.
For example in lines 342-346 we rewrite as follow:
"4.8. Antimicrobial evaluation
We performed the antimicrobial activity Larrea tridentata extracts by the Kirby-Bauer disk diffusion test. Bacteria and fungi colonies from fresh cultures were suspended in tubes containing sterile saline solution, and the turbidity of the suspensions was adjusted to match the 0.5 McFarland standard.”
Reviewer 2 Report
Comments and Suggestions for Authors
In today's scenario, drug resistance is the biggest problem in the healthcare system. To tackle this problem, we need new molecules, and this research article highlights the medicinal value of plant extracts, which is an interesting attempt. The following are the points that need to be addressed before the final acceptance:
1. In line 46, the author should write drug resistance or drug-resistant bacteria instead of bacterial resistance.
2. In line 96, delete S.aureus.
3. The authors should follow a uniform way to represent bacteria names.
4. The authors should mention the source of blood, i.e., human blood or animal blood, used in the hemolysis assay.
5. In line 291, correct “rmp” to rpm (Revolutions Per Minute). Is 100 °C temperature used for centrifugation? How did the authors maintain that high temperature for centrifugation?
6. In line 308, what does "activated" mean?
7. The authors could include images of agar plates to indicate the zone of inhibition for gram-negative and gram-positive bacteria and fungi.
8. The authors should rewrite the conclusion to make it more clear to readers.
Comments on the Quality of English Language
The language of the article needs to be improved.
Author Response
The English could be improved to more clearly express the research.
Response: we improved the English grammar for more clarity.
The introduction provide sufficient background and include all relevant references? Must be improved. Is the research design appropriate? Can be improved.
Response: we improved the background and include all relevant references as suggested.
Are the methods adequately described? Can be improved.
Response: we improved the methods as suggested.
Are the results clearly presented? Can be improved.
Response: we improved the results as suggested.
Are the conclusions supported by the results? Must be improved.
Response:
Comments and Suggestions for Authors: In today's scenario, drug resistance is the biggest problem in the healthcare system. To tackle this problem, we need new molecules, and this research article highlights the medicinal value of plant extracts, which is an interesting attempt. The following are the points that need to be addressed before the final acceptance:
1. In line 46, the author should write drug resistance or drug-resistant bacteria instead of bacterial resistance.
Response: in line 46, We changed “bacterial resistance” for “drug resistance”, as suggested.
2. In line 96, delete S.aureus.
Response: In line 96, We deleted “S. aureus”, as suggested.
3. The authors should follow a uniform way to represent bacteria names.
Response: We have consistently used microbial strain names throughout the manuscript, as suggested. Additionally, we have provided full microbial strain names to avoid confusion.
4. The authors should mention the source of blood, i.e., human blood or animal blood, used in the hemolysis assay.
Response: We have specified the source “human blood” in the hemolysis assay, as suggested.
5. In line 291, correct “rmp” to rpm (Revolutions Per Minute). Is 100 °C temperature used for centrifugation? How did the authors maintain that high temperature for centrifugation?
Response: In line 291, we changed 'rmp' to 'rpm' (Revolutions Per Minute) as suggested. We have also removed the error: 'at 100 °C.
6. In line 308, what does "activated" mean?
Response: In line 308 we have removed “activated” error to correct term “reactivated”. We have rewritten those sentences for clarity as shown below.
“The strains were preserved in nutrient media with glycerol 60 % stocks stored at -70 °C. Prior to antibacterial testing, the microorganisms were reactivated in nutrient broth and Mueller-Hinton agar (DIBICO) at 37 °C in a Thermo® incubator for 24 hours.[41].”
“reactivated' refers to the process of reanimating dormant or stored microorganisms (such as bacteria or fungi) that have been preserved in nutrient media.
7. The authors could include images of agar plates to indicate the zone of inhibition for gram-negative and gram-positive bacteria and fungi.
Response:
8. The authors should rewrite the conclusion to make it more clear to readers.
Response:
Comments on the Quality of English Language. The language of the article needs to be improved.